# On the Relevance of Soft Tissue Sarcomas Metabolic Landscape Mapping

**DOI:** 10.3390/ijms231911430

**Published:** 2022-09-28

**Authors:** Miguel Esperança-Martins, Iola F.Duarte, Mara Rodrigues, Joaquim Soares do Brito, Dolores López-Presa, Luís Costa, Isabel Fernandes, Sérgio Dias

**Affiliations:** 1Medical Oncology Department, Centro Hospitalar Universitário Lisboa Norte, 1649-028 Lisboa, Portugal; 2Vascular Biology & Cancer Microenvironment Lab, Instituto de Medicina Molecular João Lobo Antunes, Faculdade de Medicina da Universidade de Lisboa, 1649-028 Lisboa, Portugal; 3Translational Oncobiology Lab, Instituto de Medicina Molecular João Lobo Antunes, Faculdade de Medicina da Universidade de Lisboa, 1649-028 Lisboa, Portugal; 4CICECO-Aveiro Institute of Materials, Department of Chemistry, Universidade de Aveiro, 3810-193 Aveiro, Portugal; 5Orthopedics Department, Centro Hospitalar Universitário Lisboa Norte, 1649-028 Lisboa, Portugal; 6Pathology Department, Centro Hospitalar Universitário Lisboa Norte, 1649-028 Lisboa, Portugal; 7Faculdade de Medicina da Universidade de Lisboa, Clínica Universitária de Oncologia Médica, 1649-028 Lisboa, Portugal

**Keywords:** soft tissue sarcoma, single-omics, multi-omics, metabolomics, mass spectrometry, chromatography, mass spectrometry imaging, nuclear magnetic resonance

## Abstract

Soft tissue sarcomas (STS) prognosis is disappointing, with current treatment strategies being based on a “fit for all” principle and not taking distinct sarcoma subtypes specificities and genetic/metabolic differences into consideration. The paucity of precision therapies in STS reflects the shortage of studies that seek to decipher the sarcomagenesis mechanisms. There is an urge to improve STS diagnosis precision, refine STS classification criteria, and increase the capability of identifying STS prognostic biomarkers. Single-omics and multi-omics studies may play a key role on decodifying sarcomagenesis. Metabolomics provides a singular insight, either as a single-omics approach or as part of a multi-omics strategy, into the metabolic adaptations that support sarcomagenesis. Although STS metabolome is scarcely characterized, untargeted and targeted metabolomics approaches employing different data acquisition methods such as mass spectrometry (MS), MS imaging, and nuclear magnetic resonance (NMR) spectroscopy provided important information, warranting further studies. New chromatographic, MS, NMR-based, and flow cytometry-based methods will offer opportunities to therapeutically target metabolic pathways and to monitorize the response to such metabolic targeting therapies. Here we provide a comprehensive review of STS omics applications, comprising a detailed analysis of studies focused on the metabolic landscape of these tumors.

## 1. Concept and Utility of Single-Omics or Multi-Omics Strategies to Identify Biomarkers and Therapeutic Targets in Sarcomas

Sarcomas are mesenchymal malignancies of bone or soft tissues that can develop in different anatomic sites and distinct connective tissue types, showing a wide spectrum of clinical behaviors and biological activity patterns [1,2]. Sarcomas display a high level of heterogeneity, with 75% of them originating from soft tissue (STS), 15% of them being gastrointestinal stromal tumors (GIST), and 10% of them arising from bones (BS), comprising more than 170 distinct subtypes [3,4]. Sarcomas are rare neoplasms, constituting less than 1% of adult malignancies and 15% of non-hematologic childhood malignancies [3,4]. The mainstay of treatment of regional STS is surgery, while chemotherapy (CT) and radiotherapy (RT) may be used as neoadjuvant or adjuvant treatments [3]. In a metastatic setting, chemotherapy is currently the most suitable treatment choice with a palliative intent [3]. These current treatment options are based on a “fit for all” principle, not taking different sarcoma subtypes specificities and genetic differences into consideration. Their long-term success rate is also dismal, with 40% of patients with newly diagnosed sarcoma dying due to disease progression [2]. Immunotherapy and targeted therapy have progressively come under the spotlight for selected subtypes, fixing on the discrepancies that characterize each particular malignancy [5]. However, the great majority of these precision therapies stand as experimental attempts, reflecting the shortage of basic studies that strive to decodify the mechanisms and fundamental steps of sarcomagenesis.

There is, therefore, a significant urge to ameliorate sarcomas diagnosis precision, refine sarcomas clustering and classification criteria, augment the capacity of identifying sarcomas prognostic biomarkers, improve laboratory-based modeling of sarcomas, and create new hypothesis on sarcomagenesis mechanisms [2].

The swift technological development has brought single-omics and multi-omics approaches to the front line. Multi-omics merges the character of each omics approaches and examines the mechanism of tumor cells from different levels, making up for the scarcity of single-omics and giving us an integrated portray of bioactivities inside tumor cells [5]. These single and multi-omics approaches may be particularly useful to improve sarcoma knowledge at different levels.

A significant number of sarcoma single-omics studies has recently flourished, with genomics being undoubtedly the most common approach employed. However, the limited effect of loci discovered, the effect of both the coding region of genes and gene regulation, and the influence of both the environment and genetic background may explain the lack of clarifying power of these single-omics studies in regard to explaining the complete spectrum of disease development [5].

Nacev et al. recently performed the largest comparative genetic analysis of sarcomas ever reported, prospectively analyzing tumor next generation sequencing data from a cohort of 2138 sarcoma samples representing 45 histological subtypes [2]. This study identified alterations of cell cycle control and TP53, receptor tyrosine kinases/PI3K/RAS, and epigenetic regulators (such as NCOR1, particularly in uterine leiomyosarcoma (ULMS), leiomyosarcoma (LMS), and osteosarcoma (OS)) as the most common types of genetic alterations across all subtypes [2]. Besides this, several newly described associations opened new potential avenues for sarcomas precision clustering, classification, and treatment optimization, such as the association of genetic alterations of SWI/SNF chromatin remodeling complex with the development of uterine adenosarcoma, the association of TERT amplification (more frequently secondary to mutations in the TERT promoter) with the development of intimal sarcoma, the association of ATRX mutations with the development of undifferentiated pleomorphic sarcoma (UPS) and liposarcoma (LPS), together with the identification of a long upper tail of tumor mutational burden (TMB) in UPS, angiosarcoma (ANGS) and ULMS, and the transversal lack of microsatellite instability across almost all subtypes [2]. This is a paradigmatic example of a large single-omics study with potential impact on the understanding of sarcomagenesis given the size of the sample and the strength of the methodological and analytical approach.

There are also some notable examples of transcriptomic studies that led to the development of prognostic signatures using transcriptomic biomarkers such as the Complexity INdex in SARComas (CINSARC), the Genomic Grade Index (CGI), and the hypoxia-associated signatures [5].

Metabolomics offer conceptual advantages over other omics since metabolites are downstream of genes, transcripts, and proteins and any alterations in them are amplified relatively to the transcriptome or proteome. Moreover, metabolites are end-products of biochemical interactions; therefore, the metabolome may be considered as the closest biochemical representation of the phenotype [6]. Even considering these theoretical advantages, the number of metabolomics studies is smaller than the other omics studies, also typically including fewer patients and samples. However, these studies provided relevant information that helped portraying sarcoma’s metabolic landscape as it is described in detail in Section 3.

Multi-omics approaches typically combine genomics, epigenomics, transcriptomics, proteomics, and/or metabolomics as part of an advanced research strategy involving bioinformatics analysis, with a view to offer deep and dynamic understanding of disease mechanisms [5]. This type of strategy gathers mass quantity of data, evaluates, and analyzes the data with diverse mathematical methods, and acknowledges abnormal biological features with relatively small effort and time [5]. Specifically, in sarcoma research, multi-omics studies employ a great amount of data, compared with traditional sarcoma research methods, and provide, when integrated with prior knowledge of regulation pathways, a much more exact vision of mechanisms and biological activities inside sarcoma cells [5]. Importantly, deposit of multi-omics data to online databases like the Cancer Genome Atlas Program (TCGA), Gene Expression Omnibus, Expression Atlas, and Oncomine allow circumventing logistical limitations related with the low incidence rate of sarcomas, subsequent small number of cases of sarcomas to be studied and the consequent difficulty in obtaining enough omics data for an integrated analysis [5]. Indeed, many of the sarcoma multi-omics studies used information from online databases, as an addition to their data of sarcoma cases [5].

Although the pros of a multi-omics approach seem flabbergasting, there are also cons that should be taken into account. As previously mentioned, the low incidence rate of sarcoma is a major obstacle for the collection of high numbers of samples, not to mention their omics data, with the great majority of multi-omics studies using data from online databases, which is recurrently and repeatedly used (undoubtedly, a potential source of bias) [5]. The multi-omics studies that exclusively used samples from local patients often include few samples and limited data, not even reaching the statistical standards necessary for biomarker distinguishment [5]. In parallel, several studies overcome the limits imposed by insufficient data putting all STS subtypes in one analysis, independently of their genetic differences [5]. Finally, a good number of databases integrating information of different sources offer an opportunity for analysis of large numbers of samples, apart from one question: the lack of a statistical standard. The combination of data of different statistical standards may lead to conclusions with vital errors [5].

Zou et al. have published a detailed review of the different multi-omics studies in sarcomas that have been recently taking place [5].

These studies provided notable contributions to the comprehension of sarcomagenesis and to the improvement of sarcomas diagnostic accuracy. The identification of POLR3F as a diagnostic biomarker for early-stage OS, the pinpointing of the predictive potential of four genes (ENO1, TPI1, PKG1, and LDHC) and two metabolites (lactate and pyruvate) in OS, and the spotting of the central role of EWSR1-WT1 fusion gene in desmoplastic small round cell tumor (DSRCT) sarcomagenesis materialize that trend. Other good examples of these contributions include the association of early mutations in TP53, ATRX deletions, and Wnt/β-catenin alterations with the genetic diversity and differential behavior of LMS [5,7]. Moreover, the discovery of the recurrency of TP53 and histone chromatin remodeling genes mutations and the common high-expression of FGF-23 in UPS are noteworthy [5,8], as are the involvement of the concomitant hemizygous loss of genes form proximal 1p and transcriptional deregulation of the FGF8 gene in 10q24 and the regional loss of 10q and 3p and amplification of VGLL3 in neoplastic progression of myxoinflammatory fibroblastic sarcoma (MFS) [5,9]. Additionally, the finding of the presence of a superior methylation rate of the promoter regions of NPAS4 and PITX1 genes together with distinct extensive chromosomal abnormalities in uterine LMS may shed additional light on the differential characteristics between them and uterine leiomyomas [5,10]. In the same way, the higher preponderance of different copy number losses (CNL), 12q amplifications, several fusion transcripts and rearrangements of HMGA2, and CPM verified in dedifferentiated liposarcomas (DDLPS) aid to properly differentiate them from well-differentiated liposarcomas (WDLPS) [5,11]; in the same line, the more frequent amplification of regions encoding MDM2, CDK4, HMGA2 in DDLPS and the presence of specific mutations of TP53, ATRX, PTEN, and RB1 only in LMS also assist in the differentiation process of these two sarcoma types [5,12]; finally, the higher frequency of CDKN2A and CDKN2B mutations, together with the higher frequency of MYC amplifications and KDR variants in post-radiation sarcomas also aid in their distinction from sporadic ones [5,13].

Multi-omics also supplied important clues for a better definition of subtypes clustering in sarcoma—the identification of three expression subtypes of LMS (with different alterations of muscle related genes, such as DMD and MYOCD, in distinct subtypes and the association of deletions of the dystrophin gene with a worse prognosis, independently of subtype) [5,7], and the division of rhabdomyosarcoma (RMS) into two subtypes based on the existence or absence of a PAX3/7 gene fusion [5,14] are paradigmatic examples of the type of impact generated by these studies [5].

Multi-omics strategies also offered new useful insights in terms of prognostic biomarkers in sarcoma—the association of the overexpression of ribonucleotide reductase subunit M2 (RRM2) with proliferation, migration, invasion and colony formation in STS, the subsequent correlation of high expression levels of RRM2 with worse overall survival in STS patients and the arrest of cell cycle at G0/G1 phase with its inhibition make RRM2 a potential prognostic biomarker and therapeutic target in STS [5,15]. Similarly, the association of CNL of METTL4 with worse overall survival in both LMS and DDLPS makes it a potential prognostic maker for these two STS types [5,16]; higher levels of adenosine monophosphate deaminase 2 (AMPD2) expression were found to promote tumor growth and proliferation and to be associated with worse outcomes in UPS patients, making it a promising prognostic biomarker for UPS [5,17]; HMGA2 may also have a prognostic value in DDLPS [5,18], while higher levels of E-cadherin repressor Slug may also be prognostic in LMS [5,19]; ultimately, a set of seven proteins (AMPKALPHA, CHK1, S6, ARID1A, RBM15, ACETYLATUBULINLYS40, and MSH6) were also found to be related to overall survival in STS patients, while ENO1, ACVRL1, and APBB1IP also showed prognostic value in STS [5,20,21].

Finally, multi-omics approaches also provided possible novel therapeutic biomarkers, especially for targeted drugs—the correlation between the CNV of METTL4 in STS and the IC50 of 12 drugs, including Temozolomide and Olaparib, and the consequent predictable lower concentration need in clinical practice is a characteristic example [5,16]; the correlation between the homologous recombination deficiency signature found in a large proportion of LMS and the potential sensitivity of those STS to PARP inhibition [5,22], the high-frequency of CEBPA and miR-193b methylation found in DDLPS and the promising in vitro and in vivo anti-proliferative and pro-apoptotic effects of demethylating agents in DDLPS cells [5,23], the finding of the overexpression of a receptor of tyrosine kinase/RAS/PIK3CA genetic axis in RMS and the potential therapeutic benefit that may emerge from its inhibition [5,14], the mutations of SMARCB1 (SMARCB1 loss leads to aberrations in SWI/SNF complex) in epithelioid sarcoma and the promising effects of SWI/SNF in vitro inhibition in epithelioid sarcoma cells [5,24], and the promising effect of immunotherapy in sarcomas with overexpression of immune checkpoint-related IncRNAs (CD274, CD80, CD86, PCDCD1 LG2, and LGALS9), are other archetypal cases of the new insights generated by these multi-omics studies [5,25].

Interestingly, the great majority of the aforementioned studies applied and combined genomics and transcriptomics approaches, with a small number of them also employing proteomics (more commonly in combination with transcriptomics) [5]. Metabolomics was used (together with transcriptomics) in only one of the included studies [5].

Only a particularly small number of sarcoma multi-omics studies employ metabolomics techniques. This probably derives from the relative scarcity in knowledge of sarcoma metabolome, which naturally acts as a hurdle for the inclusion of a metabolomic analysis in a multi-omics study. Besides this, the technical exploration of metabolomics of all tumors is still on a very early phase when compared with the amplitude and depth of knowledge of other techniques used for the study of other omics (genomics is a good example of this) [1].

Nevertheless, it is clear that metabolomics offers an extremely rich and unique insight either as single-omics approach, or part of a multi-omics strategy, into the dynamic metabolic adaptations that support and promote sarcomagenesis, allowing a better complete and extensive comprehension of the multi-step and multi-dimensional sarcomas development process.

Methodologically, it is important to review the different techniques and approaches that have been employed with the goal of mapping and characterizing sarcomas metabolic landscape, and to summarize the resulting evidence from the application of those techniques to the current knowledge of sarcoma metabolomics.

## 2. Techniques and Approaches Used for Soft Tissue Sarcomas Metabolic Landscape Mapping

Metabolomics concerns the comprehensive measurement of metabolites in cells, tissues or biofluids, and the study of their variations in different biological conditions [26]. Metabolites are small molecules (MW < 1500 Da) that act as substrates, intermediates, and products in the complex network of reactions used by cells to generate energy and macromolecules from nutrients, among other functions. Being downstream of gene and protein activity, metabolite levels closely reflect the cells functional state and how it is affected by both intrinsic and external factors. Moreover, metabolites can act as regulatory molecules by participating in epigenetic changes, post-transcriptional and post-translational modifications, and in the allosteric regulation of protein activity [27]. Hence, characterizing the metabolome offers an invaluable window into cellular phenotypes and biological processes. Metabolomics, either alone or integrated in a multi-omics strategy, has proved particularly useful in the study of human malignancies, providing novel insights into tumorigenesis, and enabling advances in biomarker discovery and assessment of treatment responses [28].

Metabolomic analyses are usually performed either by targeted or untargeted approaches, with both types of methods requiring identification and quantification of metabolites [29].

Untargeted approaches (hypothesis-generating strategies) typically comprise the extensive assessment of available metabolites in a sample by attributing different spectral features to each metabolite, with the aid of metabolomics databases [29,30]. While many signals are usually readily assigned, a substantial number of features can remain unidentified due to the lack of spectral information for all the metabolites present and/or the high degree of data complexity and signal overlap, which make metabolite identification a laborious and time-consuming process [29,30]. Often, rather than trying to identify all the metabolites present, the focus is on identifying and quantifying those features which vary significantly between the groups compared (e.g., malignant vs. benign tissue). Then, mapping those alterations to known metabolic pathways, using biochemical knowledge and online pathway analysis tools, enables new hypotheses on molecular mechanisms to be generated.

Untargeted-discovery-global approaches permit the full scanning of the metabolome, pattern identification, and “metabolic fingerprinting” for classification of phenotypes with interacting pathways interactions [31].

Targeted or semi-targeted approaches (hypothesis-testing strategies) are based on the identification and quantification of a known specific set of metabolites in a biological sample [29,30]. Targeted approaches are performed for validation of an untargeted analysis, or when there is a priori knowledge about the metabolites of interest [31]. Regarding data acquisition methods, mass spectrometry (MS) and nuclear magnetic resonance (NMR) spectroscopy are the most frequently employed techniques [30,32].

### 2.1. MS

By measuring the mass-to-charge ratio (*m*/*z*) of ions, MS enables a wide range of ionizable molecules to be identified and quantified in complex biological samples, using very small volumes (in the order of microliters). MS is highly sensitive, being able to detect hundreds to thousands of compounds present in concentrations as low as femtomolar. Furthermore, instrumental advances and improvements in metabolite libraries have been greatly facilitating accurate metabolite identification [30,32]. This technique is generally coupled with chromatographic methods, such as liquid chromatography (LC), gas chromatography (GC), or electrophoresis, that allow the separation of compounds prior to their detection [30,32], and increase the accuracy of quantitative analysis of complex biological samples [33,34].

Some of the most significant advantages of this technique are its high sensitivity and specificity, as well as the possibility of identification of unknown metabolites [30,33].

Regarding liquid chromatography, techniques like high-performance liquid chromatography (HPLC) have offered significant improvement in the chromatographic separation, increasing resolution when compared to the standard LC techniques [30,35].

Gas chromatography is another metabolite separation technique that can be combined with MS (GC/MS) [36]. However, the latter can only cope with volatile metabolites, leading to the need of derivatizing the metabolites. This is as a disadvantage, since the derivatization process may disrupt the molecular structure of metabolites, affecting the likelihood of their posterior identification [34,36].

Although LC/MS holds a wider range of detectable metabolites, GC/MS allows a better compound identification when compared with the first [34].

Kelly et al. presented a comprehensive analysis of metabolites extracted from formalin-fixed, paraffin-embedded (FFPE) sarcoma specimens, using targeted liquid chromatography–tandem mass spectrometry (LC/MS/MS) [35]. In this study, the authors used STS samples paired with normal samples, having demonstrated excellent reproducibility and correlation between different sections of the same specimen, showing that consistent and reliable metabolomic data may be acquired from FFPE STS samples using this technique [35]. The authors observed a mean value of 106 metabolites robustly detected throughout the samples, with a trend towards higher metabolite detection rates in the tumor samples [35].

Lee et al. used a combination of LC-MS and stable isotope metabolite tracing to demonstrate that tumor cells from an UPS mouse model were relying on glutaminolysis, against from what was verified on healthy muscle cells [37]. The evidence generated underlined the role of glutamine as a donor for the TCA cycle and a nitrogen donor for aspartate production from oxaloacetate, decisively contributing towards metabolic rewiring adaptations in UPS [37].

### 2.2. MSI

Mass spectrometry imaging (MSI) is a method that enables the direct visualization and distinction of different tissue regions based on their metabolic content [30]. It requires no labeling and allows the obtention of tumor-specific molecular signatures regarding patient’s histopathologic context [30,38]. Lou et al. used this method to identify prognostic metabolite biomarkers in high grade sarcomas (osteosarcoma, leiomyosarcoma, myxofibrosarcoma, and undifferentiated pleomorphic sarcoma) using fresh frozen tumor tissue [38]. The authors identified carnitine (poor metastasis-free survival in myxofibrosarcoma patients) and inositol (1,2)-cyclic phosphate (poor overall survival in STS patients) as potential prognostic biomarkers [38].

### 2.3. NMR

NMR spectroscopy is another major analytical platform extensively employed in the field of metabolomics. The phenomenon behind NMR occurs when the nuclei of certain atoms are immersed in a static magnetic field and exposed to a second oscillating magnetic field (provided by a radiofrequencies pulse). Some nuclei experience this phenomenon, and others do not, depending on whether they possess a property called nuclear spin. NMR-active nuclei include ^1^H, ^13^C, ^31^P, and ^15^N, with ^1^H being the most observed nucleus in NMR metabolomics due to its high natural abundance [34,39]. Based on the chemical shifts, multiplicity and coupling constants of NMR-detected signals, and on their comparison to reference spectra available in databases, a wide range of metabolites can be distinguished and unequivocally identified in complex mixtures by ^1^H NMR metabolomics [39]. Moreover, as the technique is inherently quantitative, signal areas can be readily measured to assess variations in metabolite levels [39]. Other advantages of NMR include its very high reproducibility and its non-destructive nature, which allows for sample recovery for further analysis [32,34]. Moreover, NMR enables the direct analysis of intact tissues and cells, through High Resolution Magic Angle Spinning (HRMAS), providing information on the natural metabolic environment, without interference from sample extraction procedures [32,34]. On the other hand, compared to MS, ^1^H NMR is much less sensitive, being unable to detect metabolites at concentrations below a few micromolar [32,34]. This usually translates in the identification of some tens of metabolites in the ^1^H NMR profile of a biological sample, corresponding to the most abundant molecules participating in central metabolic pathways.

Metabolite identification relies on combining spectral information from one-dimensional (1D) and two-dimensional (2D) NMR experiments, such as correlation spectroscopy (COSY) and total correlation spectroscopy (TOCSY), *J*-resolved and ^1^H-^13^C correlation spectra. These experiments improve the spectral resolution and alleviate the peak overlap problem for complex biological samples, enabling unequivocal spectral assignment [32].

The most compelling feature about the use of magnetic resonance spectroscopy (MRS) in STS is the capability of MRS to go beyond anatomical and morphological information, probing the biochemistry of tissues at the cellular level. Therefore, MRS can portray the dynamic biochemical and metabolic changes that occur during the different phases of sarcoma growth and development, providing valuable information on the different steps of sarcomagenesis. The first MRS studies on STS focused on ^31^P MRS. These preliminary studies have shown important technical limitations such as the lack of localization to regions of interest, heavy contaminations with signals from muscle and poor resolution. Negendank was the first to try to overcome these issues, using ^1^H-decoupling and NOESY in conjunction with dual-tuned surface coils and accurate localization of ^31^P magnetic resonance spectra to regions of interest in three dimensions using CSI [40]. This allowed obtaining precious information about the in vivo metabolic characteristics of STS and mapping early treatment-induced metabolic changes that have shown a predictive value in terms of treatment sensitivity [40]. Subsequently, Li et al. applied the same technique to analyze 20 STS in vivo (and in vitro), having shown the power of ^31^P MRS to detect viable sarcoma cells, based on spotting prominent nucleoside triphosphates and low inorganic phosphates [41]. Moreover, inter and intra-tumoral variations of metabolite concentrations were associated with fluctuations in the relative amounts of different cell types, matrix materials, and the presence of spontaneous necrosis [41]. Jayashree et al. firstly performed 2D-correlation spectroscopy, namely, ^1^H-^1^H TOCSY, using STS samples [42]. A total of 10 STS samples and 3 samples of healthy tissue were analyzed, with levels of different lipids (phospholipid metabolites and triglycerides), choline metabolic content, and cell surface fucosylation resonance patterns allowing the distinction between benign and tumor tissue, as well as between tumor tissues of different grade [42]. By employing a proton-decoupled ^13^C MRS approach, Singer et al. have shown a correlation between the fatty acyl chain content and the histological type and grade of LPS [43]. Moreover, considering the higher detection sensitivity of ^1^H relative to ^31^P and ^13^C, ^1^H NMR analysis of LPS samples demonstrated the existence of a correlation between the degree of unsaturation of fatty acyl chains, obtained from the 2D MRS, and the mitotic activity, marker of grade or degree of differentiation, in STS (namely, LPS) [44]. The studies by Millis et al. on LPS and lipomas using 1D and 2D MRS approaches, found that NMR-visible levels of triglycerides (TG) correlated with the degree of liposarcoma differentiation, with well-differentiated tumors showing the highest TG level and the dedifferentiated and/or pleomorphic subtypes (most aggressive and metastatic) showing the lowest [45,46]. Bezabeh et al. expanded the use of MRS to other STS types, and, using 54 ex vivo healthy specimens and STS specimens of various histological types, proved that proton MRS coupled with a statistical classification strategy can differentiate between normal mesenchymal tissue from STS specimens [47]. Following the preliminary effort of evaluating metabolic characteristics of STS in vivo by Negendank, Wang et al. have shown that in vivo detection of choline by ^1^H MRS with dynamic contrast material-enhanced magnetic resonance imaging helps differentiating malignant from benign musculoskeletal tumors by revealing the presence or absence of water-soluble choline metabolites [48]. Sharma et al. deepened the understanding of the potential degree of in vitro MRS in clarifying the specific biochemical changes associated to STS in comparison to normal tissue using STS from 19 patients and performing different 2D NMR experiments [49]. They observed a significantly higher concentration of lactate in tumor tissue (indicating a higher rate of glycolysis), higher concentration of membrane metabolites glycerophosphorycholine and choline in tumor tissue (in relation with increased membrane synthesis in rapidly proliferating tumor cells), and a modification of the lipid profile in tumor tissue that correlated with sarcoma cellularity, growth rate, and differentiation [49].

More recently, new evidence has emerged highlighting the potential role of NMR spectroscopy as a predictive tool to drug sensitivity based on early metabolic adaptations of patients with STS that are treated with specific agents. Di Gialleonardo et al. used ^1^H NMR spectroscopy to study the in vitro effect of inhibiting the PI3K/mTOR pathway in cell lines, together with hyperpolarized [1-^13^C] pyruvate magnetic resonance to study the in vivo effects of the same inhibition [50]. Activation of the PI3K/mTOR pathway regulates energy production by controlling anaerobic glycolysis in cancer cells. The authors observed an overall decrease in lactate production in vitro followed by cell growth inhibition [50]. A similar quantitative reduction in lactate production accompanied by changes in tumor size was also observed in vivo, underlining the utility of ^1^H NMR spectroscopy and hyperpolarized [1-^13^C] pyruvate magnetic resonance spectroscopy in the early detection of the effects induced by pharmacological inhibition of the PI3K/mTOR pathway [50]. In parallel, Mercier et al. focused their research on desmoid tumors, aiming to identify the metabolomic differences between paired normal fibroblast and desmoid tumor cells from affected patients and normal fibroblasts from unaffected patients [51]. Desmoid tumors are locally invasive STS that lack the ability to metastasize and are often related to T41A and S45F mutations of the beta-catenin encoding gene (CTNNB1). Using untargeted NMR metabolomics, the authors highlighted differences in the metabolic profile of the two beta-catenin mutations, T41A and S45 [51]. Furthermore, administration of dasatinib and FAK inhibitor 14 resulted in a reshaped metabolic profile both in normal fibroblasts and in desmoid tumor cells, with the cell line differentiation process being led by aminoacyl-tRNA biosynthesis in mitochondria and cytoplasm, as well as by signal transduction amino acid-dependent mTORC1 activation [51]. Masiewicz et al. developed an innovative approach, employing Fast Field-Cycling, an NMR technique that measures spin-lattice relaxation rates over a wide range of magnetic fields, and that, even not being a classic metabolomic approach, provides an extraordinary insight on the molecular dynamics of the modifications of intra and extra-cellular compartments of tumor cells imposed by pathological changes in sarcomas [52]. The authors investigated spin-lattice relaxation properties of sarcoma using formaldehyde-fixed resections from surgery and analyzed the differences in the spin-lattice relaxation dispersion profiles of musculoskeletal malignancies, tissues adjacent to the tumor, and apparently healthy tissues obtained at the resection margin, with the goal of improving the diagnostic accuracy of musculoskeletal malignancies [52]. The authors identified different dispersion profiles linked with discrepancies of molecular dynamics, differences in the overall rigidity of tissues, and contrasts of water-to-protein content in different sarcomas [52]. This is particularly relevant, since, if this effect is shown in vivo, it may be used to deepen the comprehension of sarcomagenesis, to more properly assess tumor margin and to potentially increase the detection threshold for STS [52]. In some of the cases, the dispersion profiles from tumors and surrounding muscles did not differ significantly (this may be due to tumors showing similar water dynamics compared to muscles, or muscle being largely affected by the tumor and exhibiting identical dynamics), while the dispersion profiles of fat tissues clearly stand out from muscle and tumor samples, even for liposarcomas, accordingly with the typical dispersion profiles for polymer melts considering that fatty tissues in adults are mainly composed of triglycerides [52]. Fatty tissues obtained in the vicinity of liposarcomas showed significant differences in their NMR dispersion profiles when compared to those collected at the resection margins, indicating strong influences from the sarcomatous tissue in the local molecular dynamics in the neighboring tissues [52]. This is particularly important and useful since it can be an efficient contrast mechanism to delineate liposarcoma and peritumoral regions from surrounding tissues [52].

## 3. Metabolic Landscape of Soft Tissue Sarcomas

STS cells show a notable metabolic plasticity, creating interconnections between various metabolic pathways to adapt growth to the available nutrients. A prime feature frequently amplified in STS is the use of aerobic glycolysis, known as the Warburg effect, i.e., the preferential reliance on glycolysis for energy production, even in the presence of oxygen [53]. Although this is true, a radical conceptual cleavage between glycolytic and oxidative tumors must be carefully reconsidered, with hybrid phenotypes conferring more adaptable behaviors to STS cells and being, therefore, more frequent [36]. Mitochondria also finely regulate cell energetic fluxes, homeostasis and stress sensing, reactive oxygen species (ROS) production and may also command the pattern of epigenetic marking and cell differentiation, even though their role on oncogenesis of STS is still discussed [54].

There are metabolic signatures that are common to major STS subtypes. Globally in STS, RAS, PI3K, and HIPPO pathways activation typically show up coupled with a predominant glycolytic/oxidative phosphorylation (OXPHOS) signature [36,55]. The enrichment of the PGI pathway is also of notice, collectively in STS [36]. Specific signatures are also displayed by particular STS subtypes; for example, UPS shows an enrichment in PPAR/fatty acids and glycine/serine/threonine pathways while LMS features an enhanced OXPHOS signature [36].

This STS metabolic rewiring is orchestrated by oncogenic drivers upstream of growth pathways [36]. The modulation of growth factor receptors (GFR) activity be the co-engagement of restriction points controlled by tumor suppressor genes (TSG) exemplifies this concept [36,56].

Growth factor receptors (GFR) activation spark different pathways, such as RAS/MAPK and PI3K/AKT/mTOR, activity and turn on transcriptional regulators, such as JUN/FOS/EGR1, that drive cell division, promote sarcoma cell proliferation, tissue organization and trophicity, and boost sarcomagenesis [36]. The hyperactivation of RAS/MAPK and PI3K/AKT/mTOR pathways promote a Warburg effect [36]. Gain-of-function mutations of the GFR KIT or PDGFRα in gastrointestinal stromal tumor (GIST), and the activity of fusion proteins FUS-DDIT3 and PAX3-FOXO1 in myxoid liposarcoma (MLPS) and alveolar rhabdomyosarcoma (ARMS), respectively, enhance IGF1R expression, a major driver of RAS/AKT/mTOR activation, and promote sarcoma progression [36,57]. The loss of the phosphatase PTEN also promotes growth-factor independent PI3K/AKT activation, maintaining autonomous nutrient uptake in cases of LMS and malignant peripheral nerve sheath tumor (MPNST) [36,58]. RAS or PI3K/AKT-driven activation increase the expression of glucose importers (GLUT) and of the upstream ATP-consuming glycolytic enzymes hexokinase (HK) and phosphofructokinase (PFK), quantitatively regulating the glycolytic flow [36,59]. Besides this, oncogenic KRAS4A isoform and, to a lesser extent, other RAS isoforms interact with HK1 on mitochondria, evicting its allosteric inhibition by glucose-6-phosphate (G6P), enhancing glycolysis [36,60]. Either hypoxia or mutations involving the RAS pathway modulate the activity of PKM2 and PGK1, two ATP-generating enzymes in the last steps of glycolysis [36,61]. Different RAS-driven effects strengthen glycolysis over mitochondrial respiration and promote glucose- and glutamine-dependent anabolism [36].

Aerobic glycolysis is also induced in STS by alterations in the HIPPO pathway. A significant number of STS subtypes re-express genes involved in developmental pathways, such as HIPPO (that is responsible for the control of organ size) [36,62]. HIPPO engagement in intracellular adhesion complexes activates the MST and LATS kinases that phosphorylate the transcriptional factors YAP1 and TAZ, inducing their degradation [36]. On the other hand, after nuclear translocation, YAP/TAZ have synergic mitogenic effects and stimulate cell proliferation [36]. Loss of MST/LATS and/or YAP overexpression is protumoral, and this is typically verified in MCA- or radiation-induced STS [36,63] and, more specifically, UPS [36,55]. In parallel, an increased YAP/TAZ nuclear staining is a poor survival predictive characteristic in UPS, DDLPS, and embryonal rhabdomyosarcoma (ERMS) [36,64]. YAP1 is also a non-redundant oncogenic driver in MLPS (interacting with the translocation FUS-DDIT3 in the nucleus) [36,65] and a promoter of ERMS development through the transformation of satellite cells [36,66].

The HIPPO pathway is also connected to nutrient cues, namely to glucose nutritional status [36,67]. There is a well-described glucose-sensing pathway on the dependency of YAP and TAZ, whose operation is pivotal for sustaining glucose growth-promoting activity, while glycolysis is itself required to maintain YAP/TAZ tumorigenic properties, enclosing a pathological vicious cycle [15]. YAP can also alternatively promote aerobic glycolysis by a direct interaction with HIF1α in hypoxic environments [36,68]. The proglycolytic effects of YAP/TAZ engagement rely on its capability of modulating the activity of various nuclear transcriptional complexes, at different levels [36]. This explains the higher success of an STS growth-regression strategy with combined YAP/TAZ and MEK inhibition strategy rather than an isolated YAP/TAZ inhibition approach [36].

Glutamine and arginine metabolic pathways also decisively contribute to STS growth.

Circulating glutamine is one of the most abundant amino acids and an essential substrate for biosynthetic reactions. It serves as a source of carbon to refuel the TCA cycle and to produce glutathione, and it acts as a precursor for both lipid and nucleotide synthesis [69].

Different metabolomic approaches coupled with metabolite tracing imaging modalities revealed a reliance of different STS subtypes on glutaminolysis, namely, LMS (distinguishing LMS cells from healthy muscle cells) [36,37]. Mitochondrial glutaminase (GLS) plays a pivotal role in the sarcomagenesis of different STS subtypes. GLS hydrolyses glutamine to glutamate (in a process called glutaminolysis), which is subsequently dehydrogenated to α-ketoglutarate by glutamate dehydrogenase (GLUD) or an aminotransferase, promoting mitochondrial anaplerosis [36,70].

As previously mentioned, Lee et al. have proven the importance of the glutamine metabolism in UPS [37]. They have shown that different models of UPS display highly active glutamine-related metabolism and a subsequent critical dependency on glutamine [37,69]. Considering the dependency of UPS on glutamine, the therapeutic effects of a GLS inhibitor named telaglenastat (CB-839) were evaluated in vitro and in vivo, with very promising results [37,69]. The use of GLS inhibitors stopped UPS tumor growth in vitro and in vivo and are currently the object of clinical trials in humans [36].

In parallel, Lemberg et al. have shown that MPNST are also highly dependent on glutamine as MPNST cells require significantly higher glutamine concentrations post-glutamine starvation to restart their growth compared to healthy peripheral cells and immortalized Schwann cells [69,71]. They discovered that a glutamine antagonist, JHU395, significantly inhibited MPNST growth in vivo when compared to vehicle treatment [69,71]. They extensively evaluated the effect of JHU395 on glutamine-dependent biosynthesis, having analyzed, using mass spectrometry, 400 metabolites from samples of MPNST treated with JHU935 [69,71]. They found that 18 of those metabolites had a statistically significant fold change between vehicle and JHU395-treated tumors [69,71], with formylglycinamide ribonucleotide emerging as the most significantly altered metabolite [69,71].

Evidence from studies of UPS models with ^13^C or ^15^N-labeled glutamine tracer imaging methods revealed that glutamine is not only a carbon donor for the tricarboxylic acid (TCA) cycle, but also a nitrogen donor for aspartate production from oxaloacetate [36,70].

Aspartate is a pivotal carbon donor for purine and pyrimidine synthesis, perpetuating cell growth, and is also crucial for the conversion of citrulline into arginine by the activity of argininosuccinate synthase 1 (ASS1), the enzyme that starts the urea cycle [36,72]. The reaction catalyzed by ASS1 generates arginine and is crucial for the clearance of nitrogenous waste [36]. ASS1 deficiency has been verified in various STS, including MFS, due to epigenetic silencing of its promoter, leading to a dependence of cell growth on exogenous/extracellular arginine [36,73]. Re-expression of ASS1 in ASS1-deficient cancers stopped tumor growth and development [36]. The use of a pegylated arginine deiminase (ADI-PEG20) to pharmacologically deplete exogenous arginine in sarcoma cells (typically ASS1 deficient cells) resulted in prolonged arginine starvation and metabolic stress and had already been tried with promising growth restraining effects [36,73]. In fact, a short-term treatment by ADI-PEG20 applied to all LMS cell lines immediately promoted cell proliferation arrest and autophagy [36,73]. However, when these cell lines were treated with ADI-PEG20 for long periods of time, they became resistant due to the reexpression of ASS1, a regenerator of arginine [36,73]. A metabolomic profiling study of these treated cell lines showed an overall reduction in PKM2 levels, while U^13^C glucose tracing studies revealed that carbons were shifted away from lactate and citrate production and reoriented toward serine/glycine synthesis [36]. These cells growth relies less on glucose and more on OXPHOS and glutaminolysis, as an alternative source of TCA cycle intermediates via anaplerosis. A clinical trial using ADI-PEG20 in combination with Gemcitabine and Docetaxel to treat STS has been recently launched [36,73]. Concomitantly, GLS inhibition could also explore a potential synthetical lethality effect on ASS1-deficient sarcomas and is currently being studied in GIST and NF-1 mutated cancers [36].

Tryptophan (TRP) is also an essential amino acid and a small portion of its free levels in the body is employed for protein synthesis and production of certain neurotransmitters and neuromodulators [69]. The remaining majority of TRP serves a substrate for the kynurenine (KYN) pathway, one of the major catabolic routes for TRP metabolism leading to the production of NAD [69]. This reaction only happens if TRP undergoes oxidation to form KYN by the action of the enzymes indoleamine 2,3-dioxygenase 1 (IDO1), IDO2, or tryptophan-2, 3-dioxygenase (TDO2) [69]. This is a rate-limiting step for extracellular TRP depletion with multiple crucial organs such as the intestine, brain, liver, and immune cells serving as sites for TRP metabolism [69]. There is a link between IDO, cancer, and immune cell regulation, with inhibitors of this enzyme alone or in combination with immune checkpoint inhibitors being under consideration for cancer therapy [69]. Preclinical data have shown that IDO1-deficient mice are more sensitive to immune checkpoint inhibition with anti-CTLA4, anti-PD1 and anti-PDL1 agents [69,74]. TRP depletion also induces T regulatory cells proliferation, downregulates T cell receptor chain expression in CD8+ cells and induces the expression of inhibitory receptors in dendritic cells, easing tumor escape [69,75]. The safety and efficacy of the combination of pembrolizumab with metronomic cyclophosphamide was assessed in a single-arm, phase 2 multicenter study with 57 advanced STS patients (the greater part of them with LMS and UPS subtypes), showing that a significantly higher KYN to TRP plasma ratio was seen in patients during treatment [69,76]. In a follow-up preclinical study, the same group assessed the effects of KYN pathway enzymes upon PD-L1 inhibition in syngeneic models of sarcoma [69,74]. PD-L1 blockade led to an increase in enzymes involved in KYN pathway including IDO1 and IDO2 alongside inflammatory cytokines [69,74]. Finally, the authors investigated the impact of an IDO inhibitor (GDC 0919) in preclinical sarcoma models not only as a single agent, but also in combination with the anti-PD-L1 antibody. This resulted in a significant decrease in plasma KYN to TRP ratio but, pitifully, in virtually no anti-tumor activity [69,74].

The overactivity of one carbon metabolism is another metabolic feature of aggressive STS [36]. Upstream regulation of PHGDH, PSAT, PSPH, and SHMT 1/2 genes involved in serine-glycine biosynthesis, together with the overactivation of SLC1A4/5 glutamine transporter genes increase the proportion of glucose-derived 3-phosphoglycerate that is reoriented towards serine and glycine synthesis in different STS subtypes [36]. Serine and glycine can offer one carbon to tetrahydrofolate initiating the folate cycle [36]. The one-carbon metabolism ties and strengthens the knot between the folate and methionine cycles leading to augmented NAPDH and nucleotide synthesis [36,77]. Knockdown of PHGDH led to the development of a similar effect of anti-metabolite chemotherapies while dietary methionine restriction combined with radiotherapy promoted a synergic appreciable response in terms of tumor growth restriction [36].

Most of STS preferentially use particular metabolic pathways during their different developmental stages, including a global progressive increase in nucleotide synthesis [36,78].

Besides this, the use of imaging methods using PET-FDG uptake in STS patients confirmed a marked glycolytic bias in metastized and poor prognosis STS [36,79] such as ARMS [36,80]. Still, there is a significant heterogeneity within each STS subtype and between different STS subtypes, suggesting that the aerobic glycolytic phenotype might be unstable and susceptible to pharmacological control [36,81]. The level of OXPHOS varies between different tumors, but there is a general correlation between reduced mitochondrial activity, the existence of an epithelial-to-mesenchymal transition (EMT) gene signature and poor prognosis [36,78]. Specific STS exhibit higher levels of mitochondrial respiration compared to carcinomas [36,82]. The balance between glycolysis and mitochondrial respiration may be influenced by various oncogenic alterations (such as Her4 overexpression, that boosts glycolysis and glutaminolysis) and metabolic requirements [36,83]. The crosstalk between metabolic pathways may also be modified in cancer. There are some good paradigmatic examples. One of them is the expression of various phosphofructokinase (PFK) isoenzymes (such as bifunctional 6-phosphofructo-2-kinase/fructose-2,6-biphosphate (PFKB) by cancer clones as a strategy to override ATP-dependent inhibition of PFK-1 (PFK-1 is the enzyme that catalyzes the upstream reaction committing glucose to glycolysis and that is itself allosterically inhibited by high ATP levels in normal conditions) [36,84]. Conversely, an activation of a F1, 6-biphosphatase like FBP1 augments the gluconeogenic flow and restrains glycolysis [36,82]. The FBP2 isoform is often lost in STS including LPS, FS, LMS and UPS [15,85]. Augmenting FBP2 expression restrains sarcoma cell growth, through glycolysis inhibition and promotion of mitochondrial biogenesis [15,85].

Mitochondrial abnormalities also disrupt the TCA cycle, shape STS metabolic landscape and may, for example, disarray the process of transition from myoblast growth to myocyte differentiation (by the persistent activation of the classical NF-κβ activation pathway and to the induction of a pro-glycolytic HK2 isoform) leading to RMS sarcomagenesis [36,86], and promote sarcomagenesis of different STS and wild-type GIST (linked with the occurrence of mutations in TCA enzymes SDH and FH, and mutations in IDH 1 and 2) [87,88].

Tumor metabolome also influences STS progression [36]. Aerobic glycolysis induced by oncogenic or hypoxic signaling also leads to modifications in the tumor microenvironment: the lactate excretion, hypoxia-associated hypercapnia, and acidification of the extracellular milieu accelerate the degradation of the extracellular matrix and facilitate metastization [36,89]. Cancer-associated fibroblasts (CAF) have the capacity of producing lactate and 3-hydroxybutyrate that fuel cell growth, metastasis, and angiogenesis when administered to tumor-bearing mice [36,90]. Besides this, microenvironmental lactate uptake by tumors feeds their oxidative metabolism and demands the importer MCT1, a marker of mitochondrial activity and stemness in cancer and a target gene of the fusion protein ASP-SCR1/TFE3 in alveolar soft part sarcoma (ASPS) [36,91]. The sustenance of mitochondrial activity may promote metabolic plasticity, mtROS-driven anoikis, metastasis or resistance to therapy as displayed for LPS [36,92]. Metabolic plasticity, required during EMT, is still incompletely understood in poorly polarized and migration-prone mesenchymal tumor cells such as sarcoma cells [36,93]. Hybrid epithelial/mesenchymal phenotypes or shifting from E-to-N-cadherin and vimentin expression contribute to aggressiveness, metastatic properties, and drug resistance [36]. Different fusion protein events and translocations regulate epithelial differentiation in STS [36]. EMT is promoted by cytokines such as FGF, PDGFB, TGF-β that enhance glycolysis and TCA activity [36]. TGF-β signaling, for example, cooperates with the AKT and NF-κβ pathways, driving and inducing glycolysis, being also a potent antagonizer of PDK4, allowing the pyruvate entry into the TCA [36,94]. Hypoxia controls the expression of various intracellular collagen-modifying enzymes, specially OGDH enzymes that hydroxylate proline and lysine residues, contributing to the quality of collagen folding, stiffness of the tissue, and subsequently affecting cell migration [36,95]. HIF1α expression is associated with higher levels of the intracellular enzyme procollagen-lysine, 2-oxoglutarate 5-dioxygenase 2 (PLOD2) in a UPS model [36]. Loss or enhanced expression of PLOD2 abolishes or restores, respectively, the metastatic potential of HIF1α-deficient tumors and human sarcomas typically display high HIF1α and PLOD2 expression in metastatic primary lesions [36,96]. HIF can also promote ECM degradation through the induction of various metalloproteinases such as MMP or PLAUR, facilitating invasiveness [36]. Interestingly, additional features such as elevated plasmatic levels of interleukin-6 (IL-6) and tumor necrosis factor-α (TNF-α) correlate with the magnitude of tumoral cell replication [97].

The immunoreactivity of STS cells and the immune composition of STS microenvironment are relatively unexplored fields [36]. The level of infiltration of cytotoxic CD8+T cells or of myeloid cells, the expression of markers of immune-stimulation or-depression and the localization of these cells within the tumor, arise as common landmarks of tumor immunogenicity [36]. STS are portrayed as low mutational burden malignancies when compared to other cancer types and are typically considered to be poorly immunogenic and scarcely responsive to immune checkpoint blockade [36,98]. Synovial sarcoma, soft tissue, and undifferentiated LMS are the three subtypes with the lowest CNV [36,99], displaying reduced immune infiltration, while undifferentiated LPS, MPNST, alveolar sarcoma (AS), and GIST have several SCNAs, nucleotide, and chromosomal instabilities and can present high levels of lymphoid infiltration including CD8+ T cells [36]. The metabolic rewiring promoted by tumors creates a situation of competition for essential energetic resources (glucose, vitamins, and essential amino acids) leading to an impairment of immune cell functions and memory [36,99]. The exchange of fatty acids is necessary for the survival of immunosuppressive myeloid cells or T regs, especially under conditions of activation of the PI3K/AKT/mTOR axis that promotes lipogenesis [36,100]. Extracellular nucleotides released after cell death can promote immunosuppression via diverse mechanisms [36,101]. Metabolic disturbances imposed by tumor cells directly contribute to the reorganization of the microenvironment but an extensive analysis of the immune landscape in STS is still awaited [36].

## 4. Potential Impact of an Extensive Soft Tissue Sarcomas Metabolic Landscape Mapping

A substantial part of recent advances in the knowledge of STS metabolic adaptations and features is directly and intimately interlinked with the progress in data acquisition and processing methods. It is, therefore, easily predictable that the future of sarcoma metabolism exploration will continue to be fueled by technical progresses in different omics analytical modalities. There are promising examples that embody this tendency. The refinement of chromatographic and MS analyses such as Ultra High-Performance Liquid Chromatography Q-Exactive MS (UHPLC-QE MS) has permitted time to be saved in sample separation while maintaining the detection capacity of a vast spectrum of metabolites [36]. Spatially correlated analysis, mass spectrometry imaging (MALDI-MSI) can reveal to a greater extent how biomolecular ions are distributed on tissue sections, connecting their molecular identification to their spatial distribution [36]. Airflow-assisted desorption electrospray ionization mass spectrometry imaging (AFADESI-MSI) has the capability of identifying tumor-associated metabolites in situ [36]. These explorations will benefit in the future from single-cell strategies assessing the metabolic status of tumor versus surrounding cells.

Regarding this, the development of novel flow cytometry-based methods to evaluate metabolic activity, such as Met-Flow or SCENITH, is of paramount importance since these methods have proven potential to assay the metabolic status of circulating or tumor-infiltrating lymphocytes [36]. These approaches will open wide windows to fully unveil and understand the immunocyte metabolomic profiles in STS.

Developments in NMR-based methods are also expected to provide additional insights into STS. One interesting application regards the live monitoring of cellular metabolism, which is possible due to the non-destructive nature of NMR. Custom-made NMR detectors integrated with microfluidics have been recently shown to provide real-time, dynamic information on the metabolism of 3D tumor spheroids [102,103]. Moreover, spatially resolved data of 3D spheroids can also be obtained [104], enabling the distribution of nutrients and waste products to be monitored in different tumor sites.

Further detailing and more firmly drawing the lines of the metabolic panorama of different STS with the aid of more refined techniques will offer unique opportunities to therapeutically target specific metabolic pathways. This is an attractive strategy: while most tumors are genetically different, modifications in many oncogenes and TSG induce a common metabolic phenotype [69]. Targeting these shared characteristics may be considerably more profitable than treatments based solely on individualized genetic profiles [69]. Furthermore, given the strong association between metabolism and the immune microenvironment, targeting metabolic pathways to enhance immune cell function is also a growing area of research and the cornerstone of new therapies (rapamycin is a particularly good example of this strategy) [69].

Indeed, metabolomic and multi-omics studies of STS allowed the identification of several cancer cell metabolic vulnerabilities, leading to the development of distinct experimental approaches targeting glycolysis, aspartate, glutamine, or fatty-acid metabolism as it was explained previously. The GLS inhibitor telaglenastat (CB-839), the glutamine antagonist JHU395, the arginine deiminase ADI-PEG20, and the IDO inhibitor GDC 0919 are concrete examples of those approaches.

Conceptually, a key principle for metabolism-based drug development is drug specificity [105]. Several highly specific metabolic inhibitors are currently emerging, targeting catalytic and allosteric sites [105]. The concept of targeting active sites is challenged by the prevalence of hydrophobic pockets in metabolic enzymes, and, thus, allosteric inhibitors seem to offer supplementary opportunities with better specificity [105]. Active site inhibitors of LDH and allosteric GLS inhibitors are good examples of specific enzyme inhibitors, while 6-diazo-5-oxo-L-norleucine (DON), a glutamine mimetic drug that covalently binds to multiple enzymes that use glutamine, is a paradigmatic example of a multiple enzyme inhibitor [105]. There are also bold difficulties. Firstly, the capacity of neoplastic clones to constantly rewire metabolic pathways may constitute an additional challenge as they are consequently capable of developing resistance, which can be overcome by combination therapies or therapies designed to block multiple pathways instead of single-agent or single-acting therapies [105]. Secondly, the nanomolar intracellular concentrations of metabolic enzymes may pose an additional difficulty regarding the capability of generating inhibitors with effective dosage (half-maximal inhibitory concentrations in the nanomolar range is pivotal) [105]. High intracellular concentrations of enzymes needing drug doses that may cause absolute target neutralization may also be troublesome [105]. Targeted protein degradation through proteolysis targeting chimaera (PROTAC) technology—the principle is to tie up small-molecule binders to warheads that target proteins for degradation via the ubiquitin-proteasome system—might be a possible solution [104].

Inherited human disorders of metabolism could offer profitable information relative to therapeutic windows and possible side effects of targeting specific metabolic pathways with drugs [105].

A more vivid and thorough portrait of the metabolic rewiring patterns and full metabolic profiles of different STS will broaden horizons and provide a golden chance to monitorize response to this metabolic targeting therapies. Imaging methods based on metabolic tracers can be extraordinarily useful for monitoring drug activity in vivo [105]. For instance, a ^13^C-Pyruvate tracer may allow the observation of on-target effect of an LDH inhibitor, a ^18^F-glutamine tracer may permit the verification of a glutaminase inhibitor in vivo effects, while a ^11^C-acetate tracer may show the in vivo effect of the loss of acetyl-CoA synthetase short-chain family member 2 (ACSS2, that converts acetyl-CoA from acetate) [105]. A new era of in vivo, real time and high-precision functional imaging with treatment-efficacy monitoring value seems to be relatively closer.

## 5. Conclusions

Metabolomics represents a uniquely valuable approach for biomarker and therapeutic target unveiling approach in STS, either as a single-omics strategy or integrated in a multi-omics plan. STS metabolic landscape mapping will be powered by technical progresses in distinct omics analytical methods, particularly metabolomics, opening new windows for therapeutic targeting of key metabolic pathways and detailed monitorization of response to those metabolic targeting approaches.

## Data Availability

Not applicable.

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
