# Peer review of "On the Relevance of Soft Tissue Sarcomas Metabolic Landscape Mapping"

_ijms, 2022, doi:10.3390/ijms231911430_

Round 1

Reviewer 1 Report

Dear Authors

I have reviewed your paper with great interest.

I will accept your paper after a minimal revision.

My revision is:

Title: Very Good

Abstract: Very Good

Introduction and AIM: The problem and the aim are well descripting.

Materials, Patients and methods and statistics: All good.

Results: Focus on and well described.

Discussion and Thread: effectiveness Focus ON.

Soft tissue sarcomas are many common and difficult to spot, cite and discuss these papers:

Del Bravo V, Liuzza F, Perisano C, Chalidis B, Marzetti E, Colelli P, Maccauro G. Gluteal tumoral calcinosis. Hip Int. 2012 Nov-Dec;22(6):585-91. doi: 10.5301/HIP.2012.10347. PMID: 23233180.

Greco T, Cianni L, De Mauro D, Dughiero G, Bocchi MB, Cazzato G, Ragonesi G, Liuzza F, Maccauro G, Perisano C. Foot metastasis: Current knowledge. Orthop Rev (Pavia). 2020 Jun 25;12(Suppl 1):8671. doi: 10.4081/or.2020.8671. PMID: 32913603; PMCID: PMC7459381.

Prosthetic replacement with modular implants has become the most common reconstructive technique of bone loss and soft tissue sarcomas of the lower limb after tumour resection. 

Myalgia reflects generalized inflammation and cytokine response and can be the onset symptom of 78% of patients with Sarcomas. Interleukin-6 (IL-6) and tumor necrosis factor-α (TNF- α) levels in plasma correlate with the magnitude of tumoral cell replication, fever and systemic symptoms, including musculoskeletal clinical manifestations. Cite and discuss this paper:

Ripani U, Bisaccia M, Meccariello L. Dexamethasone and Nutraceutical Therapy Can Reduce the Myalgia Due to COVID-19 - a Systemic Review of the Active Substances that Can Reduce the Expression of Interlukin-6. Med Arch. 2022 Feb;76(1):66-71. doi: 10.5455/medarh.2022.76.66-71. PMID: 35422571; PMCID: PMC8976893.

References: Well chosen but to improve

Figures and Table: Very Good.

Author Response

Reviewer 1

“Dear Authors

I have reviewed your paper with great interest.

I will accept your paper after a minimal revision.

My revision is:

Title: Very Good

Abstract: Very Good 

Introduction and AIM: The problem and the aim are well descripting.

Materials, Patients and methods and statistics: All good.

Results: Focus on and well described.

Discussion and Thread: effectiveness Focus ON

References: Well chosen but to improve.

Figures and Table: Very Good”.

RESPONSE: We thank the reviewer for the kind considerations and positive message.

“Soft tissue sarcomas are many common and difficult to spot, cite and discuss these papers:

Del Bravo V, Liuzza F, Perisano C, Chalidis B, Marzetti E, Colelli P, Maccauro G. Gluteal tumoral calcinosis. Hip Int. 2012 Nov-Dec;22(6):585-91. doi: 10.5301/HIP.2012.10347. PMID: 23233180.

Greco T, Cianni L, De Mauro D, Dughiero G, Bocchi MB, Cazzato G, Ragonesi G, Liuzza F, Maccauro G, Perisano C. Foot metastasis: Current knowledge. Orthop Rev (Pavia). 2020 Jun 25;12(Suppl 1):8671. doi: 10.4081/or.2020.8671. PMID: 32913603; PMCID: PMC7459381.

Prosthetic replacement with modular implants has become the most common reconstructive technique of bone loss and soft tissue sarcomas of the lower limb after tumour resection. 

Myalgia reflects generalized inflammation and cytokine response and can be the onset symptom of 78% of patients with Sarcomas. Interleukin-6 (IL-6) and tumor necrosis factor-α (TNF- α) levels in plasma correlate with the magnitude of tumoral cell replication, fever and systemic symptoms, including musculoskeletal clinical manifestations. Cite and discuss this paper:

Ripani U, Bisaccia M, Meccariello L. Dexamethasone and Nutraceutical Therapy Can Reduce the Myalgia Due to COVID-19 - a Systemic Review of the Active Substances that Can Reduce the Expression of Interlukin-6. Med Arch. 2022 Feb;76(1):66-71. doi: 10.5455/medarh.2022.76.66-71. PMID: 35422571; PMCID: PMC8976893”.

RESPONSE: We thank and salute the reviewer for this helpful suggestion and for the time set aside to improve our manuscript. We agree with the reviewer and, therefore, added a mention to the “Dexamethasone and Nutraceutical Therapy Can Reduce the Myalgia Due to COVID-19 - a Systemic Review of the Active Substances that Can Reduce the Expression of Interlukin-6” paper, as it may be read on page 15, lines 587-588. All the subsequent references numbers were accordingly reorganized.

Reviewer 2 Report

In this manuscript, the authors reviewed the landscape of the soft tissue sarcomas previous studies in details. The manuscript is well organized with most of the previous publications summarized. This will be very important in dissecting the subtypes and metabolic differences for a precision treatment of STS in the future. The authors reviewed the current omics in biomarker and therapy related studies, updated techniques, metabolic landscape and potential impact. The authors also included an unpublished result in the review paper. Overall, the manuscript is important to improve our current understanding in STS and provide insights in potential therapeutic targets. The work is suggested to be accepted in the journal. 

However, there are some small writing errors which can be double checked before publication. For example, last paragraph in page 4, "Finallly", first paragraph in page 5 "checkpoint-related lncRNAs" should be in the same line, "MS" should have session number "2.1" ect. 

Author Response

“In this manuscript, the authors reviewed the landscape of the soft tissue sarcomas previous studies in details. The manuscript is well organized with most of the previous publications summarized. This will be very important in dissecting the subtypes and metabolic differences for a precision treatment of STS in the future. The authors reviewed the current omics in biomarker and therapy related studies, updated techniques, metabolic landscape and potential impact. The authors also included an unpublished result in the review paper. Overall, the manuscript is important to improve our current understanding in STS and provide insights in potential therapeutic targets. The work is suggested to be accepted in the journal”.

RESPONSE: We thank the reviewer for the time spent on this critical analysis of our manuscript, for these pleasant considerations and for the work improving our manuscript.

“However, there are some small writing errors which can be double checked before publication. For example, last paragraph in page 4, "Finallly", first paragraph in page 5 "checkpoint-related lncRNAs" should be in the same line, "MS" should have session number "2.1" ect. 

RESPONSE: We thank the reviewer for all these useful suggestions. Firstly, we thank the reviewer for spotting that typo and have corrected it as it may be read on page 4, line 166. Secondly, we also agree with the reviewer and arranged the text in order to put “checkpoint-related lncRNAs” in the same line, as it may be read on page 6, line 177. Thirdly, we also included numbers for each sub-section of section 2, as it may be read on page 6, line 226 for MS, page 6, line 257 for MSI, and page 7, line 266 for NMR.
